# A Study of the Associated Risk Factors for Early Failure and the Effect of Photofunctionalisation in Full-Arch Immediate Loading Treatment Based on the All-on-Four Concept

**DOI:** 10.3390/bioengineering11030223

**Published:** 2024-02-27

**Authors:** Takashi Uesugi, Yoshiaki Shimoo, Motohiro Munakata, Yu Kataoka, Daisuke Sato, Kikue Yamaguchi, Minoru Sanda, Michiya Fujimaki, Kazuhisa Nakayama, Tae Watanabe, Paulo Malo

**Affiliations:** 1Malo Dental & Medical Tokyo, Tokyo 104-0061, Japan; implant_zygoma@yahoo.co.jp (Y.S.); michiya1965@gmail.com (M.F.); kazuhisa.tokyo.okinawa@gmail.com (K.N.); tae-720114@joy.ocn.ne.jp (T.W.); implant@malodental-tokyo.com (P.M.); 2Department of Implant Dentistry, Showa University School of Dentistry, Tokyo 145-8515, Japan; munakata@dent.showa-u.ac.jp (M.M.); dsato.imp@dent.showa-u.ac.jp (D.S.); kyamaguchi@dent.showa-u.ac.jp (K.Y.); 3Department of Oral Biomaterials and Technology, Showa University School of Dentistry, Tokyo 142-8555, Japan; yu-kataoka@dent.showa-u.ac.jp; 4Department of Prosthodontics, Showa University School of Dentistry, Tokyo 145-8515, Japan; m.sanda@dent.showa-u.ac.jp

**Keywords:** all-on-four, immediate loading, early failure, complete edentulous, photofunctionalisation

## Abstract

Early implant failure occurring within 1 year after implantation has been attributed to various factors. Particularly, early failure can lead to challenges in maintaining a full-arch prosthetic device, necessitating prompt intervention, including reoperation. This study aimed to retrospectively examine implant- and patient-related factors and the effects of photofunctionalisation associated with early failure in patients who underwent treatment using the all-on-four concept in both the maxilla and mandible. We conducted this retrospective study comprising 561 patients with 2364 implants who underwent implant-supported immediate loading with fixed full-arch rehabilitation using the all-on-four concept. We aimed to assess the survival rate within 1 year after implantation and determine the risk factors influencing early failure. The 1-year survival rates after implantation were 97.1% (patient level) and 98.9% (implant level) for the maxilla and 98.5% (patient level) and 99.6% (implant level) for the mandible. There was a significant difference in the implant-level survival rates between the maxilla and mandible, with a lower rate in the maxilla (*p* = 0.043). The risk factors associated with early implant failure according to the all-on-four concept included the maxilla (implant level) and smoking (patient level). We could not find a significant effect of photofunctionalisation on early failure (*p* = 0.25) following this treatment protocol.

## 1. Introduction

Implant failure is influenced by multiple factors and categorised into early and late failure depending on the timing of implant loss. Late failure is attributed to peri-implantitis caused by bacterial infection and overloading due to bruxism or inappropriate implant design. Conversely, early failure is the result of multifaceted causes, including surgeon-related factors (skill and experience and surgical techniques) [1,2,3,4,5,6], patient-related factors (sex, age, location, bone quality, smoking, systemic diseases, and parafunction) [3,7,8,9,10,11,12,13], and implant-related factors (diameter, length, and surface characteristics) [6,7,8,9,12,14]. These factors collectively contribute to the complexity of early implant failures.

In cases of early failure, prompt intervention, including reoperation, is necessary, causing a burden on both patients and surgeons. Early failure of a single implant can lead to difficulties in maintaining the full-arch prosthesis, imposing a significant burden, particularly on patients receiving implant-supported immediate loading with fixed full-arch rehabilitation using the all-on-four concept [15,16].

Surface properties are one of the implant-related factors causing early failure [6,7,14], and in order to further improve the range of indications and predictability of implant treatment, the machined surfaces of the original Brånemark osseointegrated dental implant were improved. Compared to machined surfaces, implants with a moderate rough surface are more likely to retain fibrin matrix on the surface, facilitate osteoconductive responses and facilitate osseointegration [17,18].

In histological studies in animals, moderate implants with a moderate rough surface have higher bone-to-implant contact in the early stages of healing compared to that of machined surfaces [19,20], and in human studies, bone-to-implant contact has been found to be higher in areas with poor bone quality, such as maxillary molars [21].

Rough surfaces show higher survival rates than machined surfaces in immediate loading treatment of all edentulous situations [22], and in implant-supported immediate loading with fixed full-arch rehabilitation, rough-surfaced implants are recommended to be utilised [23].

Photofunctionalisation is a method used to further improve the surface properties of titanium implants. The effect of photofunctionalisation on titanium implants was first reported in 2009. Photofunctionalisation was shown to increase the rate of attachment, diffusion, proliferation, and differentiation of rat bone marrow-derived osteoblasts in in vitro experiments. It was also shown to increase protein adsorption capacity to titanium implants up to three times. In in vivo experiments, photofunctionalisation has been reported to promote the acquisition of osseointegration of implants and obtaining almost 100% bone-to-implant contact [24]. The effects of photofunctionalisation include alteration of physico-chemical properties and enhancement of biologic capabilities of titanium. It is defined as an overall phenomenon of modification of titanium surfaces that occurs after ultraviolet (UV) treatment [25]. It comprises the alteration of physicochemical properties, enhancement of biologic capabilities, decreasing the adhesion of wound pathogens on titanium surfaces [26,27,28], a reduction in biofilm formation on titanium surfaces [29,30], and the formation and maintenance of antimicrobial surfaces [31].

In a clinical study, it was suggested that photofunctionalisation could reduce the risk of early failure [32]. However, there is limited clinical research on its effectiveness, and no existing report has specifically assessed its impact on immediate loading implants in full-arch cases (all-on-four treatment concept). 

Despite the fact that this treatment is a very burdensome procedure, there are few reports on the risk factors affecting early implant failure, including the presence or absence of photofanctionalisation; data on real-world clinical settings are limited.

In this study, we performed a treatment using the all-on-four concept in both the upper and lower jaws. We aimed to retrospectively investigate both patient- and implant-related factors, including the presence or absence of photofunctionalisation, with respect to the survival rate (early failure) after 1 year. The objective of this retrospective study was to determine the risk factors for early implant failure of the all-on-four concept and analyse whether photofunctionalisation could contribute to reducing early implant failure.

## 2. Materials and Methods

In this retrospective study, we enrolled patients who underwent tooth extraction using all-on-four concept-based implant placement and immediate loading for fixed full-arch reconstruction at a private clinic (Malo Dental and Medical Tokyo, Tokyo, Japan) as a treatment for edentulous rehabilitation between September 2005 and October 2020. Patient selection inclusion criteria:Patients over the age of 18 of both sexes and Japanese;Severe atrophy of the maxilla or mandible in the posterior regions;Prior to treatment, a decision towards an immediately loaded implant and supported fixed complete dental prostheses had to be made;Patients had to be physically and psychologically able to tolerate conventional surgical and restorative procedures.

Exclusion criteria:


Patients who did not undergo follow-up at the private clinic;Patients with zygomatic implants;Immunosuppression;Active treatment of malignancy;Intravenous bisphosphonate therapy;Uncontrolled systemic diseases (e.g., diabetes).


This study was approved by the Ethics Committee for Research involving humans (Ethics Review Committee number: 11000688, approval number: 21-055-A), and all patients provided written informed consent.

### 2.1. Surgical Protocol

All implants used in this study had surface properties with anodic oxidation treatment (Nobel Speedy Groovy, Nobel Biocare AG, Kloten, Switzerland). Photofunctionalisation was performed just before implantation, where the implants were irradiated chairside for 1 min using the Super Osseo Integration Excimer UV system (E172-110, Excimer, Inc., Kanagawa, Japan). This equipment could generate Excimer UV radiation with a total power of 20 mW/cm^2^ and an excitation wavelength of 172 nm (Figure 1). Photofunctionalisation was performed on implant placed after August 2016, when the system was introduced.

All surgical and prosthetic procedures were performed by five surgeons including five of the authors (T.U, Y.S, M.F, K.N and T.W) with more than 10 years of clinical experience in implant dentistry. The surgical procedure according to the all-on-four concept was performed in accordance with our previously described method [33]. Surgery was performed under local anaesthesia (2% lidocaine with 1/80,000 adrenaline). The surgical and prosthetic protocol is outlined below and shown in Figure 2.

Regarding patients with remaining teeth, these teeth were extracted.A longitudinal incision was made on the mucosa distal to the first molars on both sides, and a transverse incision was made on the mucosa on the alveolar crest slightly on the lingual/palatal side to form a mucoperiosteal flap.If necessary, shaping of the alveolar bone and jawbone was performed to secure the clearance required for prosthetic device fabrication and base surface levelling.For the maxillary sinus, a portion of the anterior wall of the maxillary sinus was excised using a round burr tip, and a probe was used to explore the same area to confirm the morphology of the anterior maxillary sinus. With respect to the mandible, the mental foramen was clearly indicated, and a probe was inserted in the mesial direction along the bone surface and used to confirm the nerve running morphology.Implants (with diameter ≥ 4.0 mm) were inserted from the posterior end on both sides. The implantation position and tilt angle were determined using a standardised surgical guide (All-on-four Guide, Nobel Biocare AG, Kloten, Switzerland).The leading tip of the implant embedded in the posterior slope was placed in the mouth region equivalent to the canines while being careful to avoid causing interference. The anterior implant (with diameter ≥ 3.3 mm) was placed in the area corresponding to the middle and lateral incisors.In principle, four implant bodies were inserted. However, if it was not possible to place an implant with a length ≥ 10 mm, or if an initial fixation of ≥35 Ncm could not be obtained, additional implants should be placed nearby if necessary and possible.When inserting the implant, a straight or 17° and 30° angled abutment were attached anteriorly and posteriorly, respectively, and sutured.

To prevent postoperative infection, patients received amoxicillin (250 mg, four times daily for 5 days), 0.2% benzethonium chloride mouthwash four times daily for 2 weeks, and loxoprofen sodium (60 mg, three times daily for 5 days as an analgesic). Sutures were removed 2 weeks postoperatively.

### 2.2. Prosthetic Protocol

Briefly, impression copings were connected to each implant using an open-tray technique; thereafter, impressions and bite registrations were taken. Using the indirect method, a provisional prosthesis was fabricated with reinforced wires cast from a Co-Cr alloy into a titanium temporary cylinder and acrylic resin (PROVISTA, Sun Medical Co., Ltd., Shiga, Japan). This prosthesis was attached to the patient on the same day.

The final prosthesis was fabricated 6 months after the provisional prosthesis and was either fabricated as a titanium framework with ceramic crowns (IPS e.max Press, Ivoclar Vivadent, Schaan, Liechtenstein) or with acrylic resin crowns (anterior teeth [Bioblend, Dentsply Sirona, Bensheim, Germany] and posterior teeth [LIVDENT GRACE, GC Co., Tokyo, Japan]).

### 2.3. Follow-Up and Maintenance Protocol

The patients were instructed to maintain a soft food diet for 2–3 months postoperatively. Follow-up clinical appointments were performed at 7 days, 14 days, and 1, 2, 3, 4, 5, and 6 months. After wearing the final prosthesis, follow-up was performed every 3–6 months, during which clinical parameters were evaluated, and oral hygiene instructions were given.

### 2.4. Clinical Outcome

Early failure within 1 year of implantation:

Failure was defined as the removal of an implant within 1 year after placement owing to symptoms such as implant mobility, persistent pain, swelling, or inflammation leading to the formation of abscess. Patient-level failure was defined as failure when one or more implants were lost in a single patient.

2.Examination of risk factors affecting early failure:
Implant-related factors (photofunctionalisation, tilt, implant length, diameter, and initial fixation value);Patient-related factors (sex, smoking history (number of cigarettes: 10 cigarettes/day or more), and systemic diseases (diabetes, osteoporosis, cardiovascular diseases));Risk factors for survival rate multivariate analysis.


### 2.5. Statistical Analysis

The various risk factors influencing implant survival rate (implant failure) were analysed univariately using the chi-square test, followed by multivariate analysis using logistic regression analysis. In addition, odds ratios (OR) were calculated for the risk factors. All statistical analyses were performed using IBM SPSS Statistics 20 for Windows (International Business Machines Corp., Armonk, NY, USA). *p* < 0.05 was considered statistically significant.

## 3. Results

The study included 617 patients (345 maxilla and 272 mandible cases) with a total of 2453 implanted fixtures (1376 maxilla and 1077 mandible). The sex distribution comprised 320 male and 297 female patients, with a mean age of 56.8 ± 10.8 years. 

### 3.1. Survival Rate (Early Failure Rate) 1 Year after Implantation (Table 1)

The 1-year survival rates (early failure rates) after implantation were as follows: maxilla patient level 97.1% (2.9% early failure), maxilla implant level 98.9% (1.1% early failure), mandible patient level 98.5% (1.5% early failure), and mandible implant level 99.6% (0.4% early failure). The maxilla showed a significantly lower survival rate (higher early failure rate) at the implant level (*p* = 0.043).

### 3.2. Examination of Risk Factors Affecting Early Failure

#### Implant-Related Factors (Table 2 and Table 3)

Photofunctionalisation.

The survival rates (early failure rates) with and without photofunctionalisation were as follows: in the maxilla, at the patient level, with photofunctionalisation, it was 97.6% (2.4%) and 96.8% (3.2%) without it, while at the implant level, with photofunctionalisation, it was 99.4% (0.6%) and 98.7% (1.3%) without it. In the mandible, at the patient level, with photofunctionalisation, it was 98.8% (1.2%) and 98.4% (1.6%) without it, while at the implant level, with photofunctionalisation, it was 99.7% (0.3%) and 99.6% (0.4%) without it. Although photofunctionalisation reduced the early failure rate in both the maxilla and mandible at the patient and implant levels, no statistically significant difference was observed. The survival rates (early failure rates) with and without photofunctionalisation are summarised in Table 2.

**Table 2 bioengineering-11-00223-t002:** Early failure rate (number of implant early failures) and photofunctionalisation.

	Photofunctionalisation	Patient-Level	Implant-Level
Maxilla	Yes	2.4%(3/126 cases)	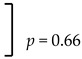	0.6%(3/486 implants)	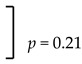
No	3.2%(7/219 cases)	1.3%(12/890 implants)
Mandible	Yes	1.2%(1/85 cases)	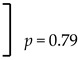	0.3%(1/333 implants)	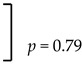
No	1.6%(184/187 cases)	0.4%(741/744 implants)

**Table 3 bioengineering-11-00223-t003:** Implant variables related to failure.

	Total (*n* = 2453)	Failure (*n* = 19)	Rate (%)	*p*-Value
Photofunctionalisation				
Yes	819	4	0.5	0.25
No	1634	15	0.9	
Implant angulation				
Straight	1229	7	0.6	
Tilted	1224	12	1	0.24
Implant length (mm)				
<10	46	0	0	―
10≤, <15	651	7	1.1	0.47
15≤, <18	712	5	0.7	0.57
18≤	1044	7	0.7	0.58
Implant diameter (mm)				
3.3	135	1	0.7	―
4.0	2291	18	0.8	0.95
5.0≤	27	0	0	0.65
Primary stability (N·cm)				
<35	143	0	0	―
35≤, <50	630	10	1.6	0.13
50≤	1680	9	0.5	0.38

Furthermore, no significant differences were found in implant angulation, implant length, implant diameter, or primary stability (*p* > 0.05).

2.Patient-related factors (Table 4).

Sex:

The failure rates were 3.4% in male and 1.0% in female patients, and thus there was a significant difference (higher in male patients, *p* = 0.043).

Smoking:

The failure rates were 4.1% in smokers and 1.3% in non-smokers, and thus there was a significant difference (higher in patient with smoking status, *p* = 0.023).

Systemic disease:

Considering systemic diseases, the study considered healthy individuals and those with diabetes, osteoporosis, and cardiovascular diseases. No significant differences were found for healthy individuals and those with diabetes and cardiovascular diseases. Although no significant difference was found for osteoporosis, there was a trend suggesting an influence on early failure (*p* = 0.051 < 0.1).

3.Risk factors for survival rate after multivariate analysis (Table 5).

To identify the risk factors for early failure, logistic regression analysis was performed for items with significant differences in implant- and patient-related factors, including photofunctionalisation. The most influential risk factor for early failure was the treatment area in the maxilla, with an OR of 3.12, and this was statistically significant after multivariate analysis (*p* = 0.0444, implant level). Smoking also showed a significant impact with an OR of 2.92, and this was confirmed after multivariate analysis (*p* = 0.0296 < 0.05, patient level). Although sex showed an OR of 1.25 in male patients, no statistical significance was observed (*p* = 0.649, patient level). Photofunctionalisation reduced early failure with an OR of 0.51. However, statistical significance was not found (*p* = 0.237, implant level).

## 4. Discussion

The survival rate of implants in the all-on-four treatment approach was favourable in the medium to long term, with the maxilla ranging between 93.9% and 100% (follow-up period: 3–18 years) and the mandible ranging between 91.7% and 100% (follow-up period: 3–18 years) [34].

Survival rate is an important indicator when considering treatment strategies. However, in a large clinical study by Derks et al. [35] on prognosis of implants in the Swedish population, the overall failure rate of implants over a 9-year observation period was 7.6%, while the failure rate of implants occurring before the superstructure attachment was 4.4% for the entire patient population. Furthermore, Lin et al. [36] conducted a study on early and late failure in 18,199 patients with 30,959 implants. They showed a high survival rate at the patient (98.0%) and implant (98.7%) levels. Among these, 194 implants (0.6%) were lost before abutment connection, and 209 implants (0.7%) were lost during the observation period of up to 6 years after abutment connection. They reported that implants lost within the first 6 months, defined as early failure, accounted for 48.1% of all implants lost during the period. Therefore, understanding and avoiding the causes of early implant failure is crucial, even more than the survival rate.

In particular, for the all-on-four treatment approach, the risk of early failure is higher because of immediate loading, and because early failure of a single implant makes it difficult to maintain the prosthesis for both jaws, the burden is significant for both patients and surgeons. The definition of early failure varies among researchers, including ① before abutment connection [2,10,12,36], ② before loading [2,4,5,6,8,11,14,32,35], ③ within 1 year of loading [3,9], and ④ within 1 year after upper structure installation [2,7], making it unclear. As this study involves immediate loading in all cases, we considered early failure within 1 year of loading for our analysis.

Maló et al. [37] reported that in cases treated with the all-on-four concept in the maxilla, there were 19 implant failures out of 968 at 5 years, with the majority occurring within 1 year of loading, accounting for 16 out of 19 (84.2%). This study determined the potential risk factors contributing to early implant failure of this all-on-four concept which was not necessarily researched and scrutinise whether the photofunctionalisation could reduce the risk of this disastrous complication.

### 4.1. Factors Related to Early Failure

#### 4.1.1. Implant-Related Factors

Early failure is mainly due to failure to achieve osseointegration; factors necessary for achieving osseointegration include implant-related factors such as length [8,9] and diameter [12,38]. However, owing to anatomical constraints, it is not possible to indefinitely use long or thick implants, and bone augmentation for implant placement also becomes a risk factor [1,3,4,5,6,8,10]. Surface characteristics, specifically roughening, improve bone-to-implant contact [39,40] and reduce the risk of early failure [6,7].

All titanium implants are covered by thin titanium oxide films. There are two types of titanium oxides: rutile and anatase. Typically, when titanium products undergo milling using lathes or machining centres under standard air conditions, they are covered by a thin film of rutile-type oxide. Consequently, traditional Branemark implants, known as machined surfaces, are enveloped by these rutile-type oxide films.

Nowadays, the most common titanium implant systems globally are TiUnite from Nobel Biocare and SLActive from Straumann. Both implant systems have moderately rough surfaces and super hydrophilicity owing to surface modifications such as anodic oxidation with discharging for TiUnite, and sandblasting, etching, and subsequent storage in a saline solution for SLActive. Employing these implant systems allows for contact osteogenesis, enhancing the rate of achievement of osseointegration and decreasing the load-free period. Understanding the biological interaction between implants and living tissue at the implant interface is crucial for developing a new generation of titanium implants aimed to achieve improved clinical outcomes.

Photofunctionalisation can be used to improve the surface properties of titanium implants.

Concerns arose regarding contamination affecting all titanium implants during storage before package opening, resulting in reduced hydrophilicity. Termed the “ageing of titanium implants”, this condition was addressed by specific irradiation using low-pressure mercury lamps emitting UV waves, which could restore the ageing process and achieve complete osseointegration. While samples treated with acid etching experienced severe contamination, conventional machined surfaces were less affected. Alongside the recommendation of UV wave devices, simpler and potentially more secure alternatives such as glow-discharging devices, UV excimer lamps, and chemical treatment involving saline solution immersion were also suggested.

Photofunctionalisation removes hydrocarbons adhering to the titanium implant surface and improves hydrophilicity, promoting osseointegration [27]. Clinically, it elevates the Implant Stability Quotient (ISQ) [41,42]. In these reports [41,42], photofunctionalisation was performed using TheraBeam Affiny (Ushio Inc., Tokyo, Japan) with a 15 min irradiation.

The device utilises fluorescent lamps with mercury to emit UV light and is widely used in the medical field. However, the Minamata Convention on Mercury, an international treaty adopted in October 2013 and enacted in 2016, imposed regulations on the production, import, and export of mercury-based products. Consequently, the use of medical products containing mercury was significantly restricted. In contrast, an excimer device generates a specific wavelength by applying a voltage to noble gas without relying on mercury. It is distinguished by its wavelength variability and emits strong light energy in the short wavelength region. Thus, an excimer-based UV irradiation device emerges as a dependable and safe alternative that does not employ mercury.

In this study, we used the Super Osseo Integration Excimer UV system (E172-110, Excimer, Inc., Kanagawa, Japan) for 1 min irradiation.

As a result, photofunctionalisation reduced the risk effect on early failure with an OR of 0.51, despite showing no statistically significant difference. However, since the implant surface used in this study was subjected to anodic oxidation, further investigation is needed to understand the impact of photofunctionalisation on different surface characteristics, such as acid etching or blast treatment. Additionally, considering the reported effect of photofunctionalisation on reducing the risk of early failure and postoperative wound dehiscence [32], further studies are warranted, particularly in cases involving bone augmentation, where wound dehiscence is more likely to occur.

In addition, as this study focused on risk factors for early failure, peri-implant bone resorption and soft tissue parameters (e.g., probing depth and bleeding) were not assessed. The effect of photofanctionalisation on peri-implant tissue should be assessed in more detail in the future, as it has been reported that photofanctionalisation could improve bone-to-implant contact [24] and create and maintain an antimicrobial surface [31].

Furthermore, in this study, maxilla treatment significantly impacted early failure at the implant level. Wu et al. [6] also reported the maxilla as a risk factor in a meta-analysis of early failure in immediate loading treatments. Considering the reports of significantly lower survival rates in the maxilla compared with the mandible in the mid to long term with treatments based on the all-on-four concept [33,43], careful monitoring is necessary for maxillary cases. The higher incidence of early failure in maxillary implants could be attributed to bone quality and factors such as the state of opposing teeth and the impact of parafunction, especially given the implementation of immediate loading therapy. Parafunction has been reported as a risk factor for early failure [13]; hence, accumulating more data for further investigation in the future is necessary.

#### 4.1.2. Patient-Related Factors

In this study, univariate analysis revealed that being male negatively impacted early failure. Similarly, several studies [6,8,9,11,35] indicated that being male may negatively influence early failure. Additionally, in treatments based on the all-on-four concept, Malo et al. observed a mid- to long-term decrease in survival rates and increased marginal bone loss associated with males in the maxilla [15]. In the mandible, an increase in mechanical complications such as prosthesis or screw fractures was associated with males [16]. This suggests that attention should be on early and late failure due to marginal bone loss and mechanical complications.

In this study, significant negative effects on early failure were observed in smokers. Similarly, many reports [3,10,11,12,13,33] indicate that smoking negatively impacts early failure. In the context of all-on-four treatments, Maló [44] reported that the survival rate of implants in smokers continued to decline even at 5 and 10 years after implantation. For smokers undergoing implant treatment, smoking cessation guidance is considered crucial for preventing early and late failure.

In this study, the relationship between the presence of systemic diseases and early failure was investigated. While no significant difference in early failure was observed based on the presence or absence of systemic diseases, osteoporosis had a slight impact on early failure (*p* = 0.051 < 0.1). Alsaadi [12] reported a negative impact of osteoporosis on early failure. In the context of treatments based on the all-on-four concept, Maló [37] reported that patients treated for osteoporosis with bisphosphonate drugs were at a higher risk of implant failure. Therefore, further investigations with an increased number of cases are warranted.

This study has some limitations. First, only one type of implant surface characteristics was considered; therefore, the impact of photofunctionalisation owing to differences in surface characteristics could not be assessed. Additionally, the effects of opposing teeth or parafunction were not investigated, and the evaluation did consider the influence of commonly used medications related to systemic diseases. In the future, we aim to accumulate more data to conduct a detailed investigation into the risk factors for early implant failure in full-arch immediate loading.

## 5. Conclusions

We conducted a multivariate analysis on the implant-related factors, patient-related factors, and the effect of photofunctionalisation on early failure within 1 year after implantation using the all-on-four approach for full-arch immediate loading treatment. The results obtained are as follows:The survival rate after 1 year of implantation was high for both the maxilla and mandible. However, at the implant level, the maxilla showed a significantly lower survival rate compared to the mandible.Among the implant-related factors, significantly higher early failure rates were seen in the maxilla.Among the patient-related factors, smoking significantly increased the risk of early failure (OR: 2.52, *p* = 0.03).Photofunctionalisation did not show the effect on early failure with an odds ratio of 0.51; however, statistical significance was not found.

## Figures and Tables

**Figure 1 bioengineering-11-00223-f001:**
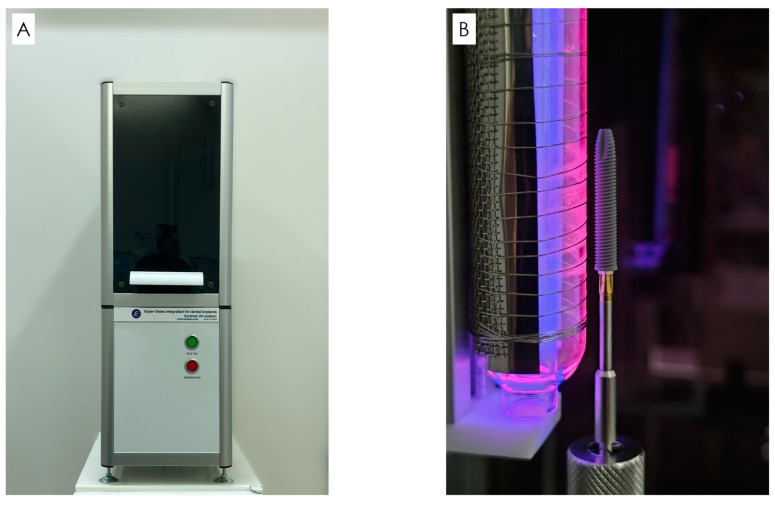
Excimer irradiation equipment. (**A**) Super Osseo Integration Excimer UV system. (**B**) Irradiation for 1 min.

**Figure 2 bioengineering-11-00223-f002:**
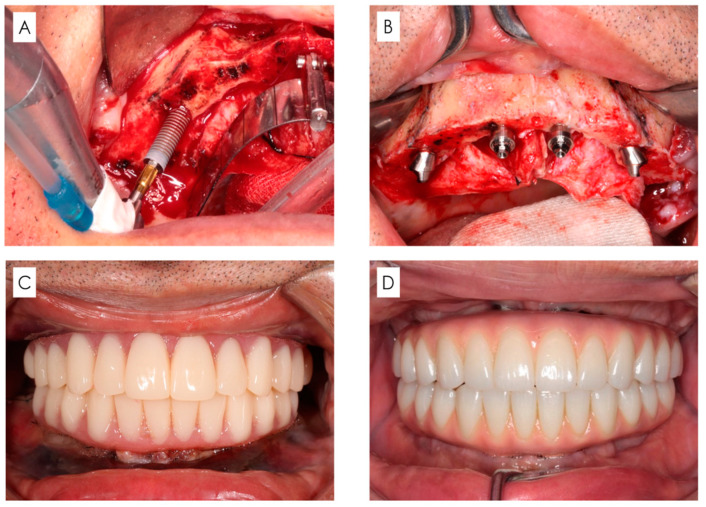
Surgical and prosthetic protocol. (**A**) Insertion of implants. (**B**) Placed implants and abutments. (**C**) Immediate loading with provisional restoration after operation. (**D**) Final restoration.

**Table 1 bioengineering-11-00223-t001:** Early failure rate (number of implant early failures).

	Patient-Level	Implant-Level
Maxilla	2.9%(10/345 cases)	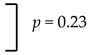	1.1%(15/1376 implants)	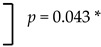
Mandible	1.5%(4/272 cases)	0.4%(4/1077 implants)

* *p* < 0.05: Chi-square test.

**Table 4 bioengineering-11-00223-t004:** Patient variables related to failure.

	Total (*n* = 617)	Failure (*n* = 14)	Rate (%)	*p*-Value
Sex				
Male	320	11	3.4	0.043 *
Female	297	3	1	
Age (years; mean ± SD)	56.8 ± 10.8			
Smoking				
Yes	220	9	4.1	0.023 *
No	397	5	1.3	
Systemic disease				
Healthy				
Yes	348	7	2	
No	269	7	2.6	0.625
Diabetes				
Yes	38	2	5.3	
No	579	12	2.1	0.2
Osteoporosis				
Yes	8	1	12.5	
No	609	13	2.1	0.051
Cardiovascular diseases				
Yes	130	3	2.3	
No	487	11	2.3	0.97

* *p* < 0.05: Chi-square test.

**Table 5 bioengineering-11-00223-t005:** Risk factors for survival rate after multivariate analysis (odds ratio).

	Risk Factor	Odds Ratio	*p*-Value
Implant-related factors for implant level			
Photofunctionalisation	Yes	0.51	0.237 > 0.05
Treatment area	Maxilla	3.12	0.0444 < 0.05 *
Patient-related factors for patient level			
Sex	Male	1.25	0.649 > 0.05
Smoker	Smoking	2.92	0.0296 < 0.05 *

* *p* < 0.05: Chi-square test.

## Data Availability

All data generated or analysed during this study are included in this published article.

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
