# Peer review of "A Study of the Associated Risk Factors for Early Failure and the Effect of Photofunctionalisation in Full-Arch Immediate Loading Treatment Based on the All-on-Four Concept"

_bioengineering, 2024, doi:10.3390/bioengineering11030223_

Round 1
Reviewer 1 Report
Comments and Suggestions for Authors
The manuscript might be of interest but it needs a major revision before publication.
Abstract
The authors should specify how many implants were photofunctionalized.
The design of the study should also be specified: is this a retrospective or prospective study?
Introduction
- The introduction section is very brief. The authors should provide a more thorough description of the background and available literature on the subject matter of their research both on the topic of full-arch immediate loading rehabilitations and on the topic of dental implant surfaces and photofunctionalization.
- The aim of the research should be clearer. Please, also specify the study design: is this a retrospective or prospective study?
Materials and Methods
- Patients’ inclusion and exclusion criteria should be specified
- It should be specified if all the implants were photofunctionalized. How many implants have been photofunctionalized? How did the authors decide which implants photofunctionalize?
- Who performed the clinical procedures? If they were performed by some of the authors, please specify it.
- Specify length and diameter of the implants inserted since only generic data are reported (i.e. diameter ≥ 4 mm: how many implants had a 4 mm diameter? How many implants had a 5 mm diameter? Etc.)
- Regarding diabetic patients: do this group include both controlled and inadequately controlled diabetes?
- The results section report data on patients’ systemic diseases. This outcome should be also specified in the materials and methods section
Discussion
- The authors should specify which is the original contribution of their research on the topic of full-arch immediate loading and dental implant photofunctionalization.
- The limitations of the present research should also be specified. One major limitation is that only early failure was investigated. The authors should discuss why they did not analyse other important clinical outcomes such as peri-implant bone resorption, probing depth, bleeding on probing, technical and biological complications, etc.
Comments on the Quality of English LanguageThe manuscript should be revised by a native English speaker.
Author Response
We thank you for taking the time and effort necessary to review our manuscript and provide us with these valuable comments and suggestions. Accordingly, we revised our manuscript and made changes to it.
Reviewer1
Thank you for your review. Corrections are highlighted in yellow marker.
Abstract
The authors should specify how many implants were photofunctionalized.
The design of the study should also be specified: is this a retrospective or prospective study?
Response: Thank you for pointing this out.
We have added a text to the abstract to indicate that this was a retrospective study.
Introduction
The introduction section is very brief. The authors should provide a more thorough description of the background and available literature on the subject matter of their research both on the topic of full-arch immediate loading rehabilitations and on the topic of dental implant surfaces and photofunctionalization.
Response: A description of the background was added to the fourth paragraph of the introduction.
The aim of the research should be clearer. Please, also specify the study design: is this a retrospective or prospective study?
Response: The objective of this study was clearly mentioned at the end of the introduction section in the revised manuscript. This was a retrospective study; this information was also added.
Materials and Methods
Patients’ inclusion and exclusion criteria should be specified
Response: The inclusion and exclusion criteria are included in the revised manuscript.
It should be specified if all the implants were photofunctionalized. How many implants have been photofunctionalized? How did the authors decide which implants photofunctionalize?
Response: The information on the presence and absence of photofunctionalisation was added to the first paragraph of subsection: 2.1.
Who performed the clinical procedures? If they were performed by some of the authors, please specify it.
Response: The revised manuscript includes the information on the practitioners who performed the procedures.
Specify length and diameter of the implants inserted since only generic data are reported (i.e. diameter ≥ 4 mm: how many implants had a 4 mm diameter? How many implants had a 5 mm diameter? Etc.)
Response: This information is presented in Table 3.
Regarding diabetic patients: do this group include both controlled and inadequately controlled diabetes?
Response: We excluded patients with inadequately controlled diabetes. This information was added to the exclusion criteria in the revised manuscript.
- The results section report data on patients’ systemic diseases. This outcome should be also specified in the materials and methods section
Response: The systemic conditions we listed in the table are also explained in the subsection 2.4 in the revised manuscript.
Discussion
The authors should specify which is the original contribution of their research on the topic of full-arch immediate loading and dental implant photofunctionalization.
Response: This point is clearly mentioned at the end of fourth paragraph of the Discussion section in the revised manuscript.
The limitations of the present research should also be specified. One major limitation is that only early failure was investigated. The authors should discuss why they did not analyse other important clinical outcomes such as peri-implant bone resorption, probing depth, bleeding on probing, technical and biological complications, etc.
Response: The limitation sentences were added to the tenth paragraph of the subsection: 4.1.1 in the revised manuscript.
In addition to the points the two reviewers mentioned, we replaced Figure 1 with newer photos. Besides, we included additional references; the number of references was also revised.
Reviewer 2 Report
Comments and Suggestions for Authors
This paper aims to evaluate retrospectively the implant-related factors, patient-related factors, and the effects of photofunctionalisation associated with early failure in patients who underwent treatment using the All-on-Four concept. It was concluded that the maxilla, sex (male),and smoking are risk factors for early failure.
Either way, I may have some comments in the various sections:
The title should also include the other factors that are studied in this work to produce an effect on the early failure not just the photofunctionalisation.
The abstract –The last 2 phrases gave similar statements of the study, please rephrase the last phrase in accordance to the conclusion section. Also, Since there were no differences in the OR determination of the effect of the photofunctionalisation is not correct to say that was a small effect.
Introduction- the introduction section needs more development in what the state of the art considers as the factors that lead to early failure of implants including major results of the vast literature on this subject. Also, photofunctionalisation as a method used to enhance the surface properties could have a more complex explanation as well as results and success of this method in in vitro, in vivo studies must be included in the introduction (since is the novelty of this paper).
Objectives- please consider including the main objectives of the study in the end of the introduction, focusing in the response included in the conclusions.
Material and Methods- inclusion criteria should be well identified. justification of the use of photofuncionalisation should be reserved to the introduction/discussion section (line 76 to 84)
Results
Review the indication of statistical significance in Tables 1 and 2.
Conclusions
Conclusion 3: Among the patient-related factors, explain how the sex (male) can significantly increase the risk of early failure if the OR was not statistically significant.
Author Response
Reviewerï¼’
Thank you for your review. Corrections are noted in yellow marker.
Thank you for your cooperation.
This paper aims to evaluate retrospectively the implant-related factors, patient-related factors, and the effects of photofunctionalisation associated with early failure in patients who underwent treatment using the All-on-Four concept. It was concluded that the maxilla, sex (male),and smoking are risk factors for early failure.
Either way, I may have some comments in the various sections:
The title should also include the other factors that are studied in this work to produce an effect on the early failure not just the photofunctionalisation.
Response: According to your comment above, the title was changed to include the risk factors for early failure of implants.
The abstract –The last 2 phrases gave similar statements of the study, please rephrase the last phrase in accordance to the conclusion section. Also, Since there were no differences in the OR determination of the effect of the photofunctionalisation is not correct to say that was a small effect.
Response: The last 2 sentences were merged due to the similarity. Also the effect of photofunctionalisation was rephrased according to your suggestions.
Introduction- the introduction section needs more development in what the state of the art considers as the factors that lead to early failure of implants including major results of the vast literature on this subject. Also, photofunctionalisation as a method used to enhance the surface properties could have a more complex explanation as well as results and success of this method in in vitro, in vivo studies must be included in the introduction (since is the novelty of this paper).
Response: The above mentioned points were included in the revised manuscript.
Objectives- please consider including the main objectives of the study in the end of the introduction, focusing in the response included in the conclusions.
Response: the objective statement was added at the end of the introduction section in the revised manuscript
Material and Methods- inclusion criteria should be well identified. justification of the use of photofuncionalisation should be reserved to the introduction/discussion section (line 76 to 84)
Response: The inclusion and exclusion criteria are listed in the revised manuscript. Sentences for photofunctionalisation were moved to the discussion section.
Results
Review the indication of statistical significance in Tables 1 and 2.
Response: Table 1 and 2 were modified in the revised manuscript.
Conclusions
Conclusion 3: Among the patient-related factors, explain how the sex (male) can significantly increase the risk of early failure if the OR was not statistically significant.
Response: Sex(male) was excluded from the risk factors for early implant failure in the revised manuscript. The introduction section was also corrected accordingly.
In addition to the points the two reviewers mentioned, we replaced Figure 1 with newer photos. Besides, we included additional references; the number of references was also revised.
Round 2
Reviewer 1 Report
Comments and Suggestions for Authors
The manuscript is of interest and in my opinion it has been improved following the reviewers suggestions, however some of the points raised in the previous review have not been fully addressed by the authors.
In particular, while the background regarding photofunctionalization has been improved, the introduction section should provide a more thorough description of the background and available literature on the topic of full-arch immediate loading rehabilitations and of dental implant surfaces.
Comments on the Quality of English Language
English language requires minor editing.
Author Response
The manuscript is of interest and in my opinion it has been improved following the reviewers suggestions, however some of the points raised in the previous review have not been fully addressed by the authors.
In particular, while the background regarding photofunctionalization has been improved, the introduction section should provide a more thorough description of the background and available literature on the topic of full-arch immediate loading rehabilitations and of dental implant surfaces.
Answer: Thank you for your advice. Additional description and associated references are included in the revised manuscript, which is highlighted with yellow. Reference numbers 18, 21 and 22 had no doi.